# Fear of Infection and the Common Good: COVID-19 and the First Italian Lockdown

**DOI:** 10.3390/ijerph182111341

**Published:** 2021-10-28

**Authors:** Lloyd Balbuena, Merylin Monaro

**Affiliations:** 1Department of Psychiatry, University of Saskatchewan, Saskatoon, SK S7N 0W8, Canada; 2Department of General Psychology, University of Padova, 35151 Padua, Italy

**Keywords:** social dilemma, fear of infection, safety measures, collective behavior, pathogens, self-control

## Abstract

In the first quarter of 2020, Italy became one of the earliest hotspots of COVID-19 infection, and the government imposed a lockdown. During the lockdown, an online survey of 2053 adults was conducted that asked about health behaviors and about the psychological and overall impact of COVID-19. The present study is a secondary analysis of that data. We hypothesized that self-control, higher socio-economic status, existing health conditions, and fear of infection were all inversely related to actions (or intentions) that violated the lockdown (i.e., infractions). Using partial least squares structural equation modeling (PLS-SEM), we found that only the fear of infection significantly dissuaded people from violating lockdown rules. Since it is not practical or ethical to sow a fear of infection, our study indicates that enacting rules and enforcing them firmly and fairly are important tools for containing the infection. This may become more important as vaccines become more widely available and people lose their fear of infection.

## 1. Introduction

In June 2020, more than 9 million people worldwide had been diagnosed with COVID-19, which resulted in 472,856 deaths [1]. Italy was an early hotspot, with infections increasing exponentially (R_0_ > 2.5) from mid-February to early March 2020 [2]. The Italian government imposed a nationwide lockdown in early March [3]. With the help of this lockdown, Italy flattened the infection curve dramatically [4].

Lockdowns have reduced the number of infections by an estimated 81 percent and have saved more than 3 million lives in 11 European countries from February to May 2020 [5]. The same report concluded that lockdowns have been the most effective government intervention by a large margin, when compared to school closures, social distancing, social isolation, and the cancelling of public events [5]. Unfortunately, lockdowns are unsustainable, and have led to the loss of millions of jobs, and economic uncertainty [6]. Lockdowns also have detrimental psychological effects, including loneliness, anxiety, depression, sleep problems, and suicidal ideation [7,8,9,10]. Feelings of isolation may have contributed to lockdown violations in both overt and covert ways.

In this work, we used the rational agent theory, studied in neoclassical economics, as a framework for understanding lockdown violations. This theory posits that individual actions are governed by the desire to satisfy needs or wants. Whatever is believed to provide the greatest satisfaction (or value) is likely to be carried out [11]. Consider somebody who is of two minds about getting a small car (which is good for the environment) and a luxury SUV (for comfort and status). Assuming that price is not a concern, the person might reason as follows: the harm to the environment is a cost that is shared by many people, while the benefit of the SUV is enjoyed solely by oneself. The person decides to buy the SUV.A COVID-19 lockdown can be viewed as a dilemma in which the common good is served by everyone’s compliance, but personal interests are maximized if everyone else complied except oneself. This is an instance of the so-called tragedy of the commons [12]. A person who shops unnecessarily gains temporary relief from confinement. Since it is impossible to police shoppers if their grocery trips are truly necessary, the common good can be undermined by self-serving actions.

We can extend the SUV vs. small car analogy to consider the role of fear. Suppose that the SUV only comes in a self-driving mode, i.e., it does not allow the person to take control of the vehicle. Although generally safe, self-driving features have been shown to fail in rare occasions, resulting in death. In this modified scenario, the imagined benefits of the SUV are tempered by the small chance of dying in an accident. It would be reasonable to infer that more risk-averse people would opt for the small car with no self-driving features. This situation mirrors the COVID-19 lockdown in which an unnecessary trip to the grocery provides relief from isolation but carries a small risk of catching the virus. People with higher anxiety are probably less likely to make unnecessary grocery trips.

We hypothesized that adherence to the lockdown was influenced by psychological traits, socio-economic status, health conditions making one more susceptible to infection, and the fear of infection. Our specific hypotheses were:*Higher self-control is inversely related to lockdown violations*. Self-control is defined as the ability to restrain impulses, and overall self-discipline [13].*Higher socio-economic status (SES) is inversely related to lockdown violations.* This was based in part on a German study that reported a positive association of higher education and engaging in COVID-19 protective measures [14].*Having health conditions is inversely related to lockdown violations.**Greater fear of infection is inversely related to lockdown violations.*

## 2. Materials and Methods

### 2.1. Participants and Data

This is a secondary data analysis of 2053 Italian adults who responded to an online survey administered in March 2020, coinciding with the first wave of the pandemic [3]. Most participants were female (*n* = 1555), 480 were male and 18 reported “other”. The respondents had a mean age (SD) of 35.81 (13.19). Please refer to the paper by Flesia et al. [3] for a complete description of the study. The materials are available on Zenodo (10.5281/zenodo.5523260). The present work did not require ethics approval, however the original study was approved by the University of Padova Ethics Committee for Psychological Research (protocol 3576, unique code 189B46FE116994F1A8D1077B835D83BB).

We calculated the adequacy of the sample size using Kock and Hadaya’s inverse square root formula [15]. A minimum of 316 people was necessary to achieve 80 percent power, at an alpha of 0.05.

### 2.2. Measures

Self-control was assessed using the 13-item Brief Self-Control Scale [16]. Linder et.al. compared unidimensional and two-factor solutions and recommended that the total score be used [17]. The internal reliability of the BSCS in this sample (Cronbach’s alpha = 0.84) was identical to that of previous studies.

Socio-economic status (SES) was assessed using participants’ typical income, their highest level of education, and how they continued to earn money during the pandemic (i.e., salary or governmental support). These indicators were based on Green’s three-item measure of socio-economic status [18]. This was chosen because of its relevance to health-related behavior and its parsimony. Since we did not have the exact job titles of respondents, we added a student status. This distinguished established workers and students from having the same attainments. This was necessary because approximately one-fourth of the respondents were students.

The fear of infection was assessed with the questions: (1) How much do you feel in danger of COVID 19 infection? (2) In the last period, are you paying more attention than usual to your physical symptoms? (3) Are you actively searching for information on the progress of the pandemic? These were Likert-type questions with five levels for the first two questions and six levels for the third. The questions were similar in content to “afraid of losing life”, “hands getting clammy”, “anxiety when watching COVID-19 news in social media” in the Fear of COVID-19 Scale [19]. The survey contained the question, *Do you currently suffer from any of the following diseases?* The available choices were: *immunosuppression, cardiovascular disease, pulmonary disease, cancer, diabetes*, and *none of the above*.

Our dependent variable was a composite of risky behaviors or intentions to disregard restrictions, which we called infractions. This was assessed with six yes-or-no questions: *(1) I respect loyally the rules imposed by ministerial ordinances, (2) I go out regularly in defiance of the ban, (3) I only go out when necessary, (4) I happened to go out for a walk in defiance of the ban, (5) I happened to go to the grocery store without real necessity, (6) I am looking for tricks to bypass the ordinances.* Questions 1 and 3 were reverse-coded to conform to the rest.

We considered self-control, SES, fear of infection and infractions as latent variables, and their respective items as indicators.

### 2.3. Analysis

We chose partial least squares structural equation modeling (PLS-SEM) to examine if infractions could be predicted by self-control, health conditions, SES, or a fear of infection. PLS-SEM was chosen because health conditions and socioeconomic status (SES) are more appropriately treated as formative variables instead of reflective variables. Reflective variables are latent constructs that are manifested by empirically measured indicators (or item responses) [20]. Covariance-based SEM (which is usually called SEM) considers underlying constructs as causes. In contrast, formative variables are defined by indicators that are assumed to be the causes of the latent variable [21]. Furthermore, covariance-based SEM requires that the indicators represent a normally distributed latent variable (or be categorized versions thereof) [22,23]. However, using polychoric correlations for ordinal indicators, for example, may still result in biased estimates and standard errors [24]. In contrast, PLS-SEM is a non-parametric method that handles non-normally distributed data, and both reflective and formative indicators [25].

To test hypotheses one to four, we regressed infractions against the four latent variables as shown in Model 1 (Figure 1). To examine if the presence of health conditions indirectly inhibited infractions by increasing the fear of infection, we added a path from health conditions to fear of infection in Model 2 (Figure 2). Confidence intervals and *p* values were calculated based on 5000 bootstrap replicates.

Appendix B Models 1 and 2 were implemented in the Stata package plssem [24] and the results were visualized, assessed for quality, and checked for consistency with SmartPLS 3 [25] and ADANCO 2.0 [26]. All three programs produced identical results.

## 3. Results

The direct effects model (Table 1 and Figure 1) shows that only fear of infection had a significant, inverse association with infractions. The other variables had an inverse association with the outcome but were not statistically significant. The indirect effect of health conditions through a fear of infection (0.04 × −0.14) was not significant (Table 2 and Figure 2). Both models had poor predictive value for infractions (R^2^ = 3.2%)

The overall fit of our two models were assessed using the standardized root mean squared residual (SRMR) [27]. SRMR quantifies the discrepancy between the correlations implied our models and the observed data [28], therefore lower values are better. The SRMRs for Models 1 and 2 were 0.69 and 0.70, respectively. These were both within the suggested cut-off value of 0.80 [29]. However, the direct-effects-only model (Model 1) was more parsimonious.

The quality of our measured constructs was assessed by inspecting the composite reliability (CR), the average variance extracted (AVE), and the possible multicollinearity. These indices were applicable only for the reflective latent variables (self-control, fear of infection, and infractions). CR is a measure of internal consistency (similar to Cronbach’s alpha) but does not require equal loading of the indicators [25]. CR values above 0.7 are preferable, although 0.60 and above are acceptable for exploratory research [25]. AVE is the mean of indicator reliabilities for a construct and should be above 0.5 [21]. (Table 3) Compared to the Fear of COVID-19 Scale which had values of 0.88 and 0.51 for CR and AVE respectively, *fear of infection* had 0.77 and 0.54. Multicollinearity is indicated by a variance inflation factor (VIF) exceeding 3.0 [21]. None of our indicators (items) were collinear, with a VIF which ranged from 1.00 to 1.76 (Appendix A Table A1).

## 4. Discussion

In a large sample of adults surveyed during the first COVID-19 lockdown in Italy, we found that only the fear of infection was inversely related to actions (or intentions) which violated government restrictions. Contrary to Hypotheses 1–3, self-control, SES, and the presence of health conditions were not related to infractions. Our results suggest that the fear of infection had a positive aspect: it dissuaded people from violating lockdown rules. Despite this, fear of infection only accounted for a minuscule amount of the outcome, so there are probably more important reasons and causes.

From the perspective of evolutionary theory, fear is an adaptive response by an organism to an external threat [30]. Avoidance is an aspect of fear that confers protection from pathogens, and can be triggered by cues such as sneezing and coughing [30]. However, it is argued that epidemics arose only when people started living in settlements [31], so there may not be an innate fear of pathogens in contrast to an innate fear of snakes [32]. This may explain why mass gatherings continued even though COVID-19 deaths and infections were constantly in the news [33]. The finding that the fear of infection promoted lockdown compliance may not have direct practical importance. Worldwide, levels of anxiety are already elevated [34], so inducing fear may simply increase psychological distress and mental health problems. Instilling a fear of infection is also ethically dubious and lacking in a theoretical basis. Clear communication of “hard truths” by the government without fear-mongering may win public trust in the long run [35]. From a policy perspective, it may be more realistic to legislate penalties appropriate to particular violations. For example, a comparison of German counties that both imposed and did not impose fines showed that fines were inversely associated with COVID-19 infection rates [36]. In effect, fines may deter rule violations. As people become accustomed to living with COVID-19, fear of infection diminishes, so financial penalties may become more relevant for health behaviors.

That greater self-control was not inversely associated with infractions is surprising. Self-control is a central concept in explaining deviant behavior. Gottfredson and Hirch postulated that criminal acts are simple, easy, and provide immediate gratification [37]. This definition of criminal acts is particularly apt for the indicators *going for a walk* and *unnecessary trip to the grocery*. According to Gottfredson and Hirch, criminals (rule violators) seek pleasure and avoid pain. People with lower levels of self-control will violate a rule when the perceived benefit exceeds the perceived cost. There is substantial (but not unequivocal) evidence that greater self-control is associated with the observance of rules, superior health, and better social adjustment [16,38]. Hence, the non-significant effect of self-control on infractions demands an explanation.

We offer three possibilities. Firstly, it is possible that the risks of COVID-19 infection may have been judged too high relative to the infractions’ rewards. This cognitive appraisal may have been influenced by the fear of infection. Although there have been previous virus outbreaks (i.e., H1N1), no previous outbreak in modern times has come close to the impact that COVID-19 has had. Secondly, a sense of solidarity (i.e., “we are all in this together “) may have also dampened self-seeking behaviors. When survival is threatened by a disaster, there can be a feeling of a shared humanity that transcends class distinctions [39]. In spite of the lockdown, people in Italy used digital resources to stay connected, and this promoted a greater sense of belonging [40]. Third, self-control during a pandemic may manifest itself more prominently in thoughts instead of actions. A Slovakian study reported that feelings of a lack of control significantly predicted the endorsement of COVID-19 conspiracy theories [41].

The nonsignificant effect of SES on infractions was also surprising. Health behaviors are influenced by personal knowledge and beliefs. A US study reported that people with a high school education (vs. a higher attainment) were less likely to intend to get vaccinated, to engage in hand-washing and masking, and to support social distancing requirements [42]. It is possible that different components of SES diverge in their relation to COVID-19 beliefs and actions. For example, among university students in Jordan, those who scored lower in a knowledge test about COVID-19 were more likely to believe in conspiracy theories [42]. Surprisingly, postgraduate students, who scored higher in the knowledge test compared to undergraduates, were more likely to violate quarantine rules [43].

The present study had several limitations. As a secondary analysis, the present study inherits the online design of the original work and its limitations [3]. Notably, older people, those with less education and with a lower SES, and men were underrepresented. With a cross-sectional design, our study cannot conclude that fear of infection causes fewer infractions. Although this is our preferred interpretation, we cannot rule out the possibility that those who had higher infractions became less afraid of infection. Among our reflective variables, *self-control* did not achieve a satisfactory AVE (Table 2). For self-control to have an AVE greater than or equal to 0.5, its indicators should have a loading of at least 0.70 [25]. Model 1 shows that only two items had at least that magnitude. One possibility is that the Brief Self-Control Scale should be divided into two factors [17]. We did not do so because these factors may represent wording effects (negative vs. positively worded items) [13]. Similarly to *self-control*, *infractions* also had unsatisfactory AVE. Importantly, *health conditions* and *infractions* were self-reported. The sensitive nature of this information may have influenced the responses obtained. Bearing these limitations in mind, our results indicate that the fear of infection served a useful purpose.

## 5. Conclusions

A higher fear of infection, but not self-control, presence of health conditions, and SES, was inversely related to self-reported violations of lockdown rules. Health conditions were not associated with fear of infection. With the increasing availability of vaccines and lockdown fatigue, the enactment of laws and their fair and firm enforcement may be needed to contain future outbreaks.

## Figures and Tables

**Figure 1 ijerph-18-11341-f001:**
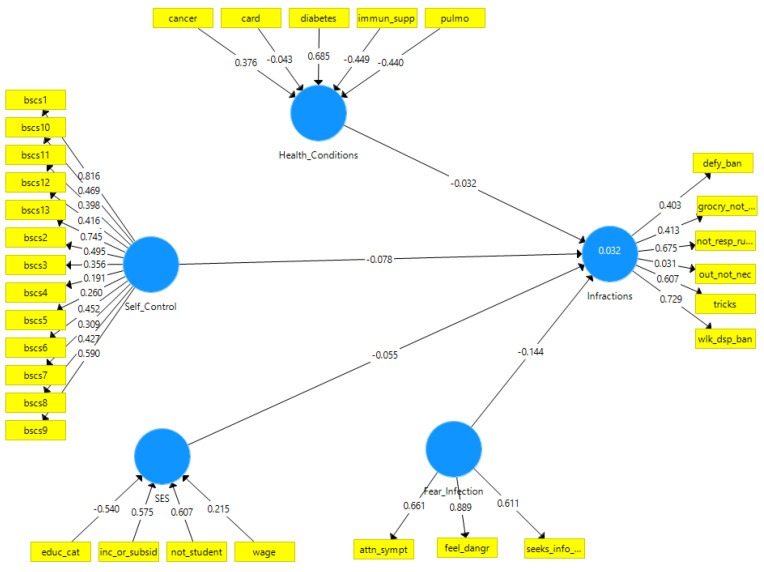
*Model 1*: Direct effects only. Please refer to Appendix A Table A1 for the exact wording of indicators. The outcome (infractions) is predicted by four latent variables indicated by circles (self-control, health conditions, SES, and fear of infection). Rectangles are the observed variables. Arrows terminating in infractions are regression coefficients. Arrows originating from a latent variable (reflective) and terminating in a rectangle represent loading. Arrows originating from a rectangle and ending in a latent variable (formative) represent weights.

**Figure 2 ijerph-18-11341-f002:**
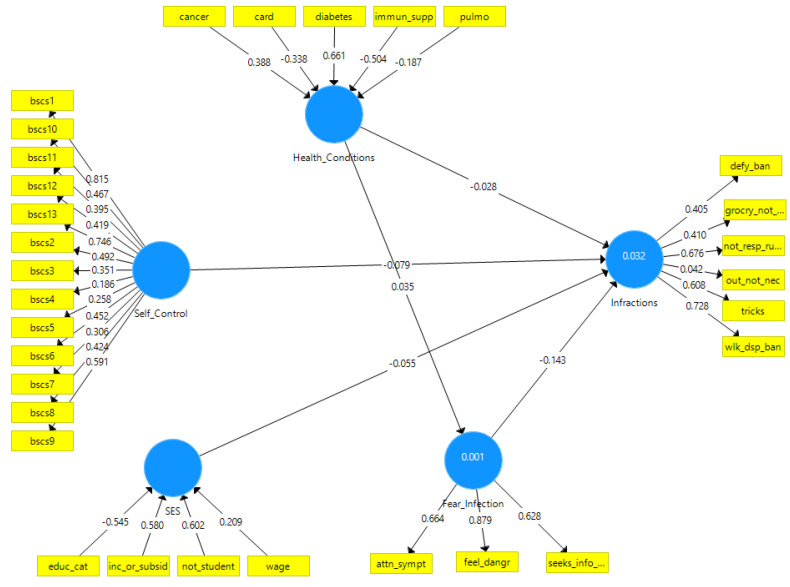
*Model 2:* Direct Effects + 1 indirect Effect. The same as Model 1 except for an added path (regression coefficient) from health conditions to fear of infection. The indirect effect of health conditions on infractions is not significant.

**Table 1 ijerph-18-11341-t001:** Model 1: Direct Effects Only.

Variable	Beta	Bootstrapped 95% CI	t	*p*
Fear of Infection	−0.14	−0.19–−0.11	−6.88	<0.001
Health Conditions	−0.03	−0.07–0.09	−0.60	0.54
SES	−0.06	−0.12–0.09	−0.80	0.43
Self-Control	−0.08	−0.15–0.12	−1.16	0.25

**Table 2 ijerph-18-11341-t002:** Model 2: Direct Effects + 1 Indirect Effect.

Variable	Beta	Bootstrapped 95% CI	t	*p*
*Direct Effects on Infractions*				
Fear of Infection	−0.14	−0.19–−0.10	−6.63	<0.001
Health Conditions	−0.03	−0.06–0.09	−0.65	0.51
SES	−0.06	−0.12–0.09	−0.79	0.43
Self-Control	−0.08	−0.15–0.12	−1.17	0.24
*Indirect Effect through Fear of Infection*				
Health Conditions	−0.01	−0.01–0.00	−0.57	0.57

**Table 3 ijerph-18-11341-t003:** Reliability of Reflective Latent Variables.

Variable	Composite Reliability	Average Variance Extracted (AVE)
Fear of Infection	0.77	0.54
Self-Control	0.78	0.24
Infractions	0.66	0.28

## Data Availability

The data are available from Zenodo (10.5281/zenodo.5523260).

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
