# Peer review of "Fear of Infection and the Common Good: COVID-19 and the First Italian Lockdown"

_ijerph, 2021, doi:10.3390/ijerph182111341_

Round 1

Reviewer 1 Report

COMMENTS

In general, the article is interesting considering that the COVID-19 pandemic requires individual actions for its control and management. And the fact that fear is relevant to comply with sanitary restrictions makes the article attractive. However, I think it is important that the authors propose and justify a theoretical model that serves as the basis and support for the empirical model. Additionally, there are other points that would help to improve the article, which I indicate below:

  • Line 57. An adequate sample is required to perform the SEM. For this reason, I suggest adding indicators on the adequacy of the sample such as the KMO Index or / and Bartlett's test. This can be done in the sample explanation or in the "Analysis" section when they indicate the model.
  • Line 65. They should include other questionnaire validation indicators and indicate the type of questions, since reability is measured with different coefficients according to the type. And if they use different types of questions (eg, yes and no; scales; among others) the least is also to include McDonald's Omega.
  • Line 74. Considering that fear is a relevant variable in the study, it is important that they include greater content validation. That is, what are they based on to say that these questions allow us to quantify fear? In this regard, you can see fear scale validation in: https://link.springer.com/article/10.1007/s11469-020-00270-8 
  • Line 127. To give more robustness to the results of the model, I suggest calculating the Fit index (RMSWEA, RMSR Kelley's criterion; or CFI, TLI).
  • Line 160. On the one hand, fear helps compliance with restrictive preventive measures. But on the other hand, fear affects people's mental health. I think this last aspect should be further discussed. In fact, there are several studies that study the determinants of fear and that can be modified or influenced through public policy. For example, see the following articles:

https://onlinelibrary.wiley.com/doi/full/10.1111/hex.13274

https://www.sciencedirect.com/science/article/pii/S0887618520300724

  • Line. 211. Authors should improve the conclusions and make them more consistent with the results of the fitted models.
  • Line 222. Considering that the study is based on a questionnaire applied to people with sensitive information, they should indicate more ethical considerations. For example, the ethics authorization code of the study with which the questionnaire was applied. Or, failing that, the authorization to use the information in the database.

Author Response

In general, the article is interesting considering that the COVID-19 pandemic requires individual actions for its control and management. And the fact that fear is relevant to comply with sanitary restrictions makes the article attractive. However, I think it is important that the authors propose and justify a theoretical model that serves as the basis and support for the empirical model.

Our theoretical model is the rational actor theory. This is related to the “tragedy of the commons” as mentioned in the Introduction. In the revision, we stated this explicitly and expanded the paragraph to serve as background for the hypothesis.

Additionally, there are other points that would help to improve the article, which I indicate below:

  • Line 57. An adequate sample is required to perform the SEM. For this reason, I suggest adding indicators on the adequacy of the sample such as the KMO Index or / and Bartlett's test. This can be done in the sample explanation or in the "Analysis" section when they indicate the model.

We agree with the need to justify the adequacy of our sample size. However, the model we are using is PLS-SEM, because of the exploratory nature of the analysis and because our data are not multivariate normal.

KMO index and Bartlett’s test are appropriate for covariance-based SEM. We followed Kock and Hadaya’s inverse square root method rule. We therefore added a paragraph stating this under Participants and Data

“We calculated the adequacy of the sample size using Kock and Hadaya’s inverse square root formula (Ref 14). A minimum of 316 people was necessary to achieve 80 percent power at an alpha of .05.”

  • Line 65. They should include other questionnaire validation indicators and indicate the type of questions, since reability is measured with different coefficients according to the type. And if they use different types of questions (eg, yes and no; scales; among others) the least is also to include McDonald's Omega.

McDonald’s omega could not be calculated for Self-Control because this scale does not meet the homogeneity requirement.  This may be related to findings that the Brief Self Control Scale has two factors (Lindner article, Ref 16).  Fear and Infractions have binary indicators so omega would not be appropriate. For these reasons, the indicators we are using are Composite Reliability and Average Variance Extracted. We explained these in the last paragraph of page 5. Note that the Fear of COVID-19 scale suggested by the reviewer (next point) also uses Composite Reliability and Average Variance Extracted as reliability indices.

  • Line 74. Considering that fear is a relevant variable in the study, it is important that they include greater content validation. That is, what are they based on to say that these questions allow us to quantify fear? In this regard, you can see fear scale validation in: https://link.springer.com/article/10.1007/s11469-020-00270-8.

We added a sentence in page 3 that compared the content of our questions with those of the Fear of COVID-19 scale.

The questions were similar in content to “afraid of losing life”, ”hands getting clammy”, “anxiety when watching COVID-19 news in social media” in the Fear of COVID-19 scale (Ref 18).

We also compared the CR and AVE of our ad hoc scale with the Fear of COVID-19 scale in the last paragraph of page 5.

  • Line 127. To give more robustness to the results of the model, I suggest calculating the Fit index (RMSWEA, RMSR Kelley's criterion; or CFI, TLI).

We used SRMR because this is the fit index that is appropriate for models in PLS-SEM. We added a paragraph that described what SRMR represents and the values for Models 1 and 2. (page 5, second to the last paragraph)

  • Line 160. On the one hand, fear helps compliance with restrictive preventive measures. But on the other hand, fear affects people's mental health. I think this last aspect should be further discussed. In fact, there are several studies that study the determinants of fear and that can be modified or influenced through public policy. For example, see the following articles:

https://onlinelibrary.wiley.com/doi/full/10.1111/hex.13274

https://www.sciencedirect.com/science/article/pii/S0887618520300724

The articles suggested by the reviewer identify predictors of fear: personality traits, having loved ones who are at risk, sensational media messages, younger age, and lower education.  We believe that these are beyond the control of public authorities.  What the government can do is to provide a clear message.

We added the sentence, “Clear communication of “hard truths” by the government without fear-mongering may win public trust in the long run [34].

  • 211. Authors should improve the conclusions and make them more consistent with the results of the fitted models.

 As suggested, we reported our main findings in the Conclusion before stating our recommendation.

  • Line 222. Considering that the study is based on a questionnaire applied to people with sensitive information, they should indicate more ethical considerations. For example, the ethics authorization code of the study with which the questionnaire was applied. Or, failing that, the authorization to use the information in the database.

Under participants and data, we added a sentence that specified the authorization code of the original study. 

Reviewer 2 Report

I think the results presented in the article are very interesting. However, I have some comments on the manuscript.

Introduction

I would advise you to expand on this part a bit. The aim of this article should be particularly clearly stated.

I was a bit surprised by the references next to the hypotheses. Maybe it would be better to add them in the footnotes?

Methods

I think the information about what kind of sample is necessary, even though it is a secondary analysis. I checked and this is the limitation of the primary study. 

Line 79: „The respondents were asked about health conditions that increased the risk of 79 COVID-19 infection.” This fragment was unclear for me. I know that you put the hypothesis “Having health conditions is inversely related to lockdown violations”, but this fragment should clearly states that respondents were asked if they have specyfic health conditions.

Discussion

You have to improve limitation section! Your research inherits the limitations of primary research. You should clearly admit the limitation of online sample.

Conclusion

This is a recommendation rather than a conclusion.

Author Response

I think the results presented in the article are very interesting. However, I have some comments on the manuscript.

Introduction

I would advise you to expand on this part a bit. The aim of this article should be particularly clearly stated.

We expanded this section by describing the rational economic agent theory as our basis for why people would commit infractions (or intend to do so). We hope that this part motivates the hypotheses being presented.

I was a bit surprised by the references next to the hypotheses. Maybe it would be better to add them in the footnotes?

In this version, we formatted our hypotheses in italic font. The sentences following the italicized sentences provide clarification—these are what the references are pertaining to. We hope that this addresses the concern.

Methods

I think the information about what kind of sample is necessary, even though it is a secondary analysis. I checked and this is the limitation of the primary study. 

We acknowledge the online nature of the study as a limitation. Please refer to our response below (under Discussion) regarding limitations.

Line 79: „The respondents were asked about health conditions that increased the risk of 79 COVID-19 infection.” This fragment was unclear for me. I know that you put the hypothesis “Having health conditions is inversely related to lockdown violations”, but this fragment should clearly states that respondents were asked if they have specyfic health conditions.

We clarified this in the following way—

“The survey contained the question, Do you currently suffer from any of the following diseases? The available choices were: immunosuppression, cardiovascular disease, pulmonary disease, cancer, diabetes, and none of the above.”

Discussion

You have to improve limitation section! Your research inherits the limitations of primary research. You should clearly admit the limitation of online sample.

We acknowledged this by stating—

“As a secondary analysis, the present study inherits the online design of the original work and its limitations [3]. Notably, older people, less educated, lower SES, and male sex were underrepresented.”

We also added--

“Importantly, health conditions and infractions were self-reported. The sensitive nature of this information may have influenced the responses obtained.”

Conclusion

This is a recommendation rather than a conclusion.

This was also pointed out by the other reviewer. We now include our two main findings in the conclusion.